# Co-Delivery of Therapeutics and Bioactive Gas Using a Novel Liposomal Platform for Enhanced Treatment of Acute Arterial Injury

**DOI:** 10.3390/biom13050861

**Published:** 2023-05-19

**Authors:** Shao-Ling Huang, Melanie R. Moody, Xing Yin, David D. McPherson, Hyunggun Kim

**Affiliations:** 1Division of Cardiology, Department of Internal Medicine, McGovern Medical School, The University of Texas Health Science Center at Houston, Houston, TX 77030, USA; melanie.r.moody@uth.tmc.edu (M.R.M.); xing.yin@uth.tmc.edu (X.Y.); david.d.mcpherson@uth.tmc.edu (D.D.M.); 2Department of Biomechatronic Engineering, Sungkyunkwan University, Suwon 16419, Republic of Korea

**Keywords:** liposome, PPARγ agonist, rosiglitazone, drug delivery, atherosclerosis

## Abstract

Atherosclerosis is a complex, multi-stage disease characterized by pathological changes across the vascular wall. Endothelial dysfunction, inflammation, hypoxia, and vascular smooth muscle cell proliferation contribute to its progression. An effective strategy capable of delivering pleiotropic treatment to the vascular wall is essential to limit neointimal formation. Echogenic liposomes (ELIP), which can encapsulate bioactive gases and therapeutic agents, have the potential to deliver enhanced penetration and treatment efficacy for atherosclerosis. In this study, liposomes loaded with nitric oxide (NO) and rosiglitazone, a peroxisome proliferator-activated receptor agonist, were prepared using hydration, sonication, freeze-thawing, and pressurization. The efficacy of this delivery system was evaluated in a rabbit model of acute arterial injury induced by balloon injury to the common carotid artery. Intra-arterial administration of rosiglitazone/NO co-encapsulated liposomes (R/NO-ELIP) immediately following injury resulted in reduced intimal thickening after 14 days. The anti-inflammatory and anti-proliferative effects of the co-delivery system were investigated. These liposomes were echogenic, enabling ultrasound imaging to assess their distribution and delivery. R/NO-ELIP delivery exhibited a greater attenuation (88 ± 15%) of intimal proliferation when compared to NO-ELIP (75 ± 13%) or R-ELIP (51 ± 6%) delivery alone. The study demonstrates the potential of echogenic liposomes as a promising platform for ultrasound imaging and therapeutic delivery.

## 1. Introduction

Pathologic and physiologic studies have demonstrated that atherosclerosis is a diffuse disease [1]. The intima undergoes deformation and protrudes outward during atheroma formation, along with neointimal hyperplasia, which involves the proliferation of vascular smooth muscle cells (VSMCs), clot formation, and inflammation. For atheroma treatment, a strategy with pleiotropic actions is more likely to be effective in limiting smooth muscle cell proliferation and inflammation [2].

Peroxisome proliferator-activated receptor γ (PPARγ) is a nuclear hormone receptor which inhibits atheroma progression in both animals and humans [3,4]. Several prospective studies have shown that PPARγ agonists stimulate the production of endothelial nitric oxide synthase (eNOS) in human umbilical vein endothelial cells (HUVEC) [5,6] and increase the number and function of endothelial progenitor cells in patients with coronary artery disease [7]. PPARγ agonists have been found to reduce neointima formation after balloon injury in rats [8] and in-stent restenosis in atherosclerotic rabbits [9]. Furthermore, they decrease the thickening of intimal and medial complexes in carotid arteries and in-stent restenosis after coronary intervention in patients [10]. However, the clinical usefulness of these treatments has been restricted due to their systemic effects [11,12]. A solution to this problem may lie in developing localized delivery methods that can enhance therapeutic permeability into the vascular wall and maximize the therapeutic benefits of treating atherosclerosis.

Two types of carriers are utilized for the local delivery of drugs: nonbiodegradable carriers, including polymethylmethacrylate beads, and biodegradable carriers, such as polylactic polymers and liposomes [13,14,15]. Liposomes, which are vesicles consisting of a phospholipid bilayer that encloses an aqueous compartment, exhibit biodegradability and non-toxicity and are capable of encapsulating hydrophilic or hydrophobic compounds within either the aqueous compartment or the phospholipid bilayer. Liposomes offer several advantages, including their small molecular size, which can be decreased even further through techniques such as sonication or filtering. These liposome/lipid complexes can exhibit a broad range of sizes from 20 nm to 10 μm and may exist in either unilamellar or multilamellar forms.

The local administration of therapeutic agents is challenged by the transfer of these agents across the endothelium from the bloodstream to the surrounding tissue. The diffusion and transendothelial permeability of the arterial wall are critical determinants of the penetration of these agents, which are influenced by their molecular weight, hydrophobic/hydrophilic properties, and electrostatic charge [16,17]. The suboptimal performance of therapeutic agent delivery mechanisms for large molecules may be attributable, in part, to the inability of the tissue to uptake these therapeutics at the target site in satisfactory concentrations within a short period, consequently hindering the desired therapeutic effects [18,19].

In order to improve the transport and uptake of therapeutic agents during vascular injury, we have developed nitric oxide-containing echogenic liposomes (NO-ELIP) and demonstrated the ability of this technology to enhance the penetration of therapeutic carriers into the arterial wall. Our hypothesis suggests that the co-encapsulation of NO with PPARγ agonists in ELIP can augment the penetration of PPARγ agonists into the vascular wall. In addition, NO provides endothelial protection due to its anti-inflammatory, antithrombotic, and antiproliferative properties [20,21]. Co-administration of exogenous NO in combination with a PPARγ agonist has the potential to provide supplemental NO to the arterial wall and enhance therapeutic effects.

The present study employs rosiglitazone as a prototype of PPARγ agonists. Rosiglitazone belongs to the family of PPARγ agonists. The developed delivery strategies for rosiglitazone are also relevant for other glitazones. The use of innovative therapeutic delivery platforms in combination with these drugs may offer a promising approach for attenuating atherosclerosis.

## 2. Materials and Methods

### 2.1. Co-Encapsulation of NO and Rosiglitazone into Echogenic Liposomes

Liposomes were composed of 1,2-dipalmitoyl-*sn*-glycero-3-ethylphosphocholine (EDPPC), 1,2-dioleoyl-*sn*-glycero-3-phosphocholine (DOPC, Avanti Polar Lipids, Alabaster, AL, USA), and cholesterol (CH, Avanti Polar Lipids, Alabaster, AL, USA) at a mole ratio of 60:30:10. First, 5 mg of lipids were mixed in chloroform. To prepare rosiglitazone-containing liposomes (R-ELIP) and rosiglitazone/NO-containing liposomes (R/NO-ELIP), 600 μg of rosiglitazone was dissolved in ethanol and added to the mixture. The solvent was evaporated using argon (Ar) in a 50 °C water bath to form a thin film on the wall of a glass vial. The lipid film was subjected to vacuum (<100 mTorr) for 4–6 h to completely remove the solvent. Next, the dried lipid film was hydrated with 0.32 M deoxygenated mannitol (500 μL), resulting in the formation of liposomes. The liposomes were then centrifuged at a low speed (1000× *g*) for 5 min to separate any unencapsulated rosiglitazone. Due to its very low solubility, free rosiglitazone remained in the pellet after centrifugation. The supernatant containing rosiglitazone-encapsulated liposomes (R-ELIP) was transferred into a sealed 2-mL glass vial with a Teflon-rubber septum cap. Moreover, in order to prepare NO-ELIP and R/NO-ELIP, NO (Specialty Gases of America, Inc., Toledo, OH, USA) was washed and deoxygenated with a saturated NaOH solution. Using a 12-mL syringe attached to a 27 G × ½” needle, 10-mL of a gaseous mixture of NO and Ar (1:9) was injected into the glass vial through the Teflon-rubber septum. Previous experiments conducted under the same conditions did not observe any gas leakage within 24 h. The NO/Ar gas-pressurized liposomal or rosiglitazone-encapsulated liposomal dispersion was frozen at −70 °C and then thawed by releasing the pressure in the vial upon removing the cap. After thawing, the gas encapsulation, release profile, and delivery characteristics of NO-containing liposomes (NO-ELIP), rosiglitazone-containing liposomes (R-ELIP), and rosiglitazone/NO-containing liposomes (R/NO-ELIP) were studied. The volume of encapsulated gas in the liposomes was determined using the syringe method as described previously [22].

### 2.2. Encapsulation and Release of Rosiglitazone from Liposomes

A spectrofluorometer (Bio Tek Synergy H1, microplate reader, Agilent, Santa Clara, CA, USA) was employed to measure the fluorescence intensity of rosiglitazone. Fluorescence was detected at an excitation wavelength of 247 nm and an emission wavelength of 367 nm. A standard curve was obtained by measuring the fluorescence intensity of a series of rosiglitazone concentrations, which was subsequently used to determine the concentration of unknown samples.

The encapsulation efficiency of rosiglitazone was assessed by centrifugation-based separation of the free rosiglitazone from the encapsulated form. The release kinetics of rosiglitazone were determined using serial dialysis under sink conditions. Specifically, 600 µL of R-ELIP and R/NO-ELIP were loaded into a dialysis tube (300 kDa MWCO, 0.79 mL/cm, Spectrum Laboratories Inc., Breda, The Netherlands). The dialysis tube was placed into 20 mL of acceptor buffer, which contained 20 mg/100 mL beta-cyclodextrin, 280 mM sodium chloride, and 25 mM HEPBS (N-(2-Hydroxyethyl)piperazine-N′-(4-butanesulfonic acid) at pH 8.4. At predetermined intervals of 0, 10, 20, 30, 40, 60, 120, 240 (for R-ELIP), 360, and 480 min, and at 24, 48, and 84 h (for R/NO-ELIP), 100 µL samples were obtained from the acceptor buffer and replaced with the same volume of fresh solution. The rosiglitazone concentration in the receiving solution at each time point was measured using a fluorometer with excitation and emission wavelengths of 247 nm and 367 nm, respectively. At the end of the experiment (84 h), triton x-100 was added to destroy the liposomes and the fluorescent intensity was determined.

The encapsulation efficiency was calculated using the following formula:% encapsulation = F_supernatant_/F_total_ × 100 (%)(1)

### 2.3. Ultrasound Imaging of R/NO-ELIP

The velocity of ultrasound propagation is higher in liquid media than in gas media due to the differences in acoustic impedance, which leads to changes in acoustic reflectivity that can be visualized through ultrasound imaging. In this study, an intravascular ultrasound (IVUS) catheter was utilized to demonstrate the presence of gas encapsulation in liposomes. The liposome samples, which included conventional empty liposomes, NO-ELIP, and R/NO-ELIP, were diluted to 25 µg lipids/mL and placed in 12 × 16-mm glass vials for analysis. Utilizing a 15-MHz high-frequency IVUS imaging catheter, the samples were imaged with a 15-MHz high-frequency intravascular ultrasound image catheter, as previously described [23]. Images were recorded and subsequently digitalized, allowing for the assessment of mean grayscale values (MGSV) across the entire image on a scale of 0–256.

### 2.4. Smooth Muscle Cell Culture and Proliferation Assay

Primary rat vascular smooth muscle cells (VSMCs) were cultured in Dulbecco modified Eagle medium (DMEM; GIBCO-BRL, Grand Island, New York) supplemented with 10% fetal bovine serum (FBS), 100 U/mL penicillin, and 100 mg/mL streptomycin in 75-cm^2^ flasks at 37 °C under 5% CO_2_ and 95% air. Only cells between passages 5 and 8 were used in the experiments. VSMCs were seeded onto 48-well tissue culture plates and cultured in DMEM with 10% FBS. Once the cells reached 80% confluence, the medium was replaced with a solution containing DMEM and 0.3% FBS, and the cells were serum-starved for 24 h. Quiescent cells were then stimulated with 10% FBS, and the dose-dependent effect of R-ELIP on VSMC proliferation was evaluated by treating stimulated cells with 1, 2, 5, 10, 20, 30, and 40 μg/mL R-ELIP. After 24 h of treatment, a fresh culture medium DMEM supplemented with MTT(3-(4,5-Dimethylthiazol-2-yl)-2,5-Diphenyltetrazolium Bromide, 1.2 mM) was added and incubated at 37 °C for 4 h to label the cells. Subsequently, dimethyl sulfoxide (DMSO) was added to each well, and the VSMC proliferation was quantified using a microplate reader (SpectraMax M5, Molecular Devices, San Jose, CA, USA) at a wavelength of 550 nm, with a reference wavelength of 650 nm. All experiments were conducted using quiescent cells at passages 4–10.

### 2.5. Endothelial Cell Culture and Western Blot Assay

Human umbilical vein endothelial cells (HUVECs) were cultured in 75-cm^2^ flasks in EGM™-2 BulletKit™ (Lonza, Switzerland) at 37 °C under 5% CO_2_ and 95% air until passages 2 to 6. Subsequently, the cells were seeded onto 48-well tissue culture plates and treated with varying doses (0, 10, 20, 30, and 40 μg/mL) of R-ELIP to determine its dose dependence on PPAR expression. At 24 h of treatment, the cells were washed with phosphate buffer saline (PBS) and lysed with cell lysis buffer (20 mmole/L Tris, 100 mmole/L NaCl, 10% triton X-100, 1 mmole/L EDTA, 1 mmole/L sodium orthovanadate, 2.5 mmole/L sodium pyrophosphate, 0.5% sodium deoxycholate, 1X protease inhibitor; Sigma-Aldrich, St. Louis, MO, USA). After electrophoresis, the separated molecules were transferred to polyvinylidenedifluoride (PVDF) immune-blot membranes (Bio-Rad, Hercules, CA, USA), which were then blotted with mouse anti-PPARγ primary antibody (1:2000, Abcam, Cambridge, MA, USA) and secondary antibody (Abcam, Cambridge, MA, USA). The blots were re-probed with β-actin to confirm equal loading. Intercellular adhesion molecule (ICAM) expression was detected using enhanced chemiluminescence-based ECL plus detection kits (Pierce, Rockford, IL, USA).

### 2.6. Evaluation of Endothelial Permeability

The impact of NO-ELIP on endothelial permeability was evaluated using rabbit aortas ex vivo. The aortas were treated with 100 µg of NO-ELIP for 2 min and compared to the control group. To determine the impact of NO-ELIP treatment on endothelial permeability, fluorescein-labeled molecules with varying sizes, including FITC-dextran (MW 9600), FITC-BSA (MW 66,000), and FITC-liposomes (300 nm in size), were administered to the aortas and incubated for 10 min. Following incubation, the aortas were washed, snap-frozen, and subjected to histological sectioning and fluorescent microscopy analysis.

### 2.7. Animal Studies for Local Delivery of R/NO-ELIP

The Animal Welfare Committee at the University of Texas Health Science Center in Houston approved all animal experiments. A total of 21 male New Zealand White rabbits, weighing 3.0–4.5 kg, were used in this study. The animals were fed an atherogenic diet containing 0.2% cholesterol and 4% coconut oil for two weeks prior to balloon denudation. On the day of surgery, the animals were anesthetized using ketamine (35 mg/kg) and xylazine (5 mg/kg) in conjunction with 1–3% isoflurane. A 2F Fogarty catheter (Edwards Lifesciences, Irvine, CA, USA) was utilized to perform balloon injury to the common carotid artery. A sheath was inserted into the right external carotid artery, and a catheter was introduced through the sheath into the common carotid artery. The balloon was inflated with 200 µL of saline solution and passed through the common carotid artery three times over a distance of 3 cm. Following balloon injury, local delivery of NO-ELIP, R-ELIP, or R/NO-ELIP was administered into the distally occluded common carotid artery through the sheath and allowed to dwell for 2 min. Following the occlusion suture release and sheath removal, the external carotid artery was ligated, and the wound was closed, enabling the rabbit to recover. Controls included sham-operated carotid arteries, balloon-injured carotid arteries without treatment, and contralateral common carotid arteries without injury or treatment.

The animals were maintained on a high-cholesterol diet and sacrificed 2 weeks post-surgery. The carotid arteries were dissected, sectioned into 2.5 mm segments, and immersed in 4% formalin dissolved in PBS for 24 h. The fixed tissues were then embedded in paraffin, and histological evaluation was performed using H&E staining. All sections from the injured segment of each artery were examined in a blind fashion using Image-Pro Plus (Media Cybernetics, Bethesda, MD, USA). Cross-sectional area measurements were conducted for the lumen and the areas encompassed by the internal and external elastic laminae. To determine the intimal cross-sectional area of the carotid artery segments, the area of the lumen was subtracted from the area bounded by the internal elastic lamina. Similarly, the medial area was determined by subtracting the area enclosed by the internal elastic lamina from the area encompassed by the external elastic lamina.

### 2.8. Statistical Analysis

The statistical analysis for intergroup comparisons was conducted using a *t*-test or one-way analysis of variance (ANOVA) with SigmaStat (Version 3.5, Systat Software Inc., Point Richmond, CA, USA) and Statistica (Version 8.0, StatSoft Inc., Tulsa, OK, USA) software. Multiple group differences were assessed utilizing the post-hoc Tukey HSD test for unequal N. A *p*-value less than 0.05 was considered statistically significant. Data were presented as mean ± SEM.

## 3. Results

### 3.1. Co-Encapsulation and Release of NO and Rosiglitazone

The encapsulation efficiency of rosiglitazone in R/NO-ELIP was 83 ± 2 (SD)%. Co-encapsulation of 100 ± 6 µg rosiglitazone with 0.045 μmol NO in 1 mg liposomes was observed. The release kinetics of rosiglitazone from R/NO-ELIP demonstrated a faster release of rosiglitazone within one hour (over 25% of the encapsulated rosiglitazone released within 4 h, followed by a slower release over 48 h (Figure 1A).

NO has been shown to have therapeutic effects at low doses but can become toxic at high doses. We have demonstrated 0.045 μmol of NO encapsulation in 1 mg of liposomes without any observed side effects [24]. This encapsulation method also rendered the liposomes echogenic, enabling ultrasound imaging (Figure 1B–E). The release of NO from R/NO-ELIP was evaluated by measuring the echogenicity of the liposomes. A gradual release of NO from R/NO-ELIP was found over 60 min (Figure 1F).

Given the NO release from R/NO-ELIP within the first hour, we hypothesized that the NO release might impact the release of rosiglitazone from R/NO-ELIP. To test this hypothesis, the first 4-h release kinetics of rosiglitazone from R/NO-ELIP were compared with that of R-ELIP. Although a trend of faster rosiglitazone release from R/NO-ELIP was observed in the first hour, no significant difference was found (Figure 1A).

### 3.2. Effect of R/NO-ELIP on Smooth Muscle Cell Proliferation

Quiescent VSMCs were stimulated by 10% FBS and exposed to varying concentrations of rosiglitazone. FBS was found to promote VSMC proliferation, while rosiglitazone treatment inhibited serum-stimulated growth in a dose-dependent manner (Figure 2A), with an EC_50_ value of 12.96 μg/mL (Figure 2B).

### 3.3. Effect of R/NO-ELIP on Endothelial Cell Inflammation

Cultured HUVECs were exposed to varying concentrations of rosiglitazone to investigate the effects on PPARγ expression. Western blot analysis revealed that R-ELIP increases PPARγ in a dose-dependent manner in these cells (Figure 3A), with an EC50 of 23.46 μg/mL (Figure 3B).

### 3.4. NO-Enhanced Molecular Penetration into the Vascular Wall

The effect of NO on enhancing molecular penetration into the vascular wall was investigated. Enhanced endothelial permeability to molecules of varying sizes was observed following treatment with NO-ELIP (Figure 4A). NO-ELIP facilitated molecular penetration into the vascular wall, leading to a 2.63-fold increase in FITC-Dextran (MW 9600), a 2.14-fold increase in FITC-BSA (MW 66,000), and a 7.2-fold increase in FITC-ELIP (size 300 nm) (Figure 4B).

### 3.5. Effect of R/NO-ELIP on Intimal Hyperplasia

The combination of a high-cholesterol diet and balloon denudation of the carotid arteries resulted in extensive neointimal hyperplasia and luminal narrowing when compared to uninjured arteries (Figure 5A). The effect of R/NO-ELIP on neointimal hyperplasia was investigated and compared to other treatments. R/NO-ELIP delivery exhibited a greater attenuation (88 ± 15%) of intimal proliferation when compared to NO-ELIP (75 ± 13%) or R-ELIP (51 ± 6%) delivery alone (Figure 5B–E).

## 4. Discussion

In this study, we have developed a liposomal formulation that allows the co-delivery of water-insoluble drugs and bioactive gas for the localized combination therapy of atheroma. Our previous research has developed a unique freeze-under-pressure method for loading NO gas into liposomes, which has been effective in inhibiting intimal hyperplasia [22]. However, the administration of bioactive gases is challenging due to the lack of suitable methods and potential adverse effects. One of the methods for systemic administration is gas inhalation, which is inefficient and has systemic side effects. To overcome these problems, liposomes can be used as a gas carrier due to their hydrophilic and hydrophobic compartments that can encapsulate agents with different biochemical properties [25,26,27]. Similar to our liposomal formulation, lipids can entrap air and other gases for ultrasound-reflective enhancement, and a range of lipid-based contrast agents are commercially available for diagnostic applications [28,29]. The encapsulation of air is presumed to behave similarly to hydrophobic drugs, with the gas residing between the two monolayers of the liposome bilayer or as a monolayer-covered air bubble within the aqueous compartment.

Neointimal hyperplasia is characterized by a range of pathological changes, including vascular endothelial dysfunction, smooth muscle cell proliferation, and inflammation, which are observed in both endogenous atheroma and in-stent restenosis post-stent implantation. To address this condition, we propose that carriers capable of delivering both NO and anti-inflammatory agents may represent a promising therapeutic strategy. By improving the previously developed NO encapsulation technique, we have successfully co-encapsulated rosiglitazone into NO-ELIP. Our preparation method resulted in the co-encapsulation of 100 ± 6 µg rosiglitazone and 0.045 μmol NO within 1 mg of liposomes. The observed release kinetics of rosiglitazone from R/NO-ELIP extended for up to 48 h. Due to the hydrophobic nature of the rosiglitazone [30], it is plausible that rosiglitazone remained localized within the lipid bilayer when co-encapsulated alongside NO (Figure 6). Importantly, our data indicated the retention of NO co-encapsulation, as demonstrated by the sustained echogenicity of the ELIP.

To address the potential issue of drug permeation through the dialysis membrane acting as a rate-limiting step in the assays, a larger pore-size membrane with a molecular weight cutoff (MWCO) of 300 kDa was utilized. Additionally, cyclodextrin was added to the acceptor solution to increase the solubility of rosiglitazone and facilitate its passage through the membrane. Cyclodextrins are cyclic oligosaccharides that possess a hydrophobic cavity capable of forming inclusion complexes with hydrophobic drugs such as rosiglitazone. Despite a 30-min delay in rosiglitazone detection in the acceptor buffer when compared to the release rate in the absence of the membrane, this approach confirmed that R/NO-ELIP exhibited a faster initial release followed by a slower release over 48 h (Figure 1A). The delay in drug detection also affected the R-ELIP control group. The release kinetics of rosiglitazone from both liposomal formulations were evaluated by this approach.

In order to investigate the therapeutic potential of the newly developed liposomal co-encapsulation and co-delivery systems for cardiovascular disease, a carotid endothelial injury was induced in Wistar rats. The animals were treated with either NO-ELIP, R-ELIP, or R/NO-ELIP. The neointimal cross-sectional areas of the animals treated with R/NO-ELIP were significantly lower than those treated with NO-ELIP alone or R-ELIP alone. The mechanism underlying this effect is likely related to the action of NO, which facilitated the penetration of R/NO-ELIP into the vascular wall.

Therapeutic penetration into the vascular wall is enhanced by the combination of PPARγ agonists and NO, resulting in increased vascular protection. This mechanism may be partially attributed to NO. PPARγ expression is found in various vascular cell types, including endothelial cells, monocytes/macrophages, and both the medial and intimal VSMCs. For optimal efficacy, therapeutic agents should penetrate all layers of the atheroma. We have demonstrated that NO promotes molecular penetration into the vascular wall across various sizes. Thus, our liposomal delivery platform provides a combined delivery of NO and PPARγ agonists, which enhances therapeutic penetration and inhibits inflammation and vascular proliferation.

Pharmacologically, PPARγ modulates the proliferation of VSMCs by directly influencing the protein kinase G-dependent (PKG) pathway and reduces neointimal hyperplasia following vascular injury [31]. Interestingly, the impact of rosiglitazone on PKG is independent of NO or cyclic guanosine monophosphate (cGMP), which are PKG upstream modulators. In addition, activation of PPARγ exerts inhibitory effects on adhesion cascades and vascular inflammatory events, with a distinct regulatory role in atheroma physiology [32]. PPAR expression is observed in vascular endothelial cells and other vascular cells [33]. By negatively regulating the signaling pathways of nuclear factor kappa B (NF-κB) and activator protein-1, PPARα suppresses the expression of inflammatory genes, including vascular cell adhesion molecule-1 (VCAM-1) [34]. In this study, we investigated the effects of rosiglitazone on PPAR phosphorylation in cultured HUVECs. Our findings demonstrate a dose-dependent increase in PPAR phosphorylation.

The release kinetics of NO from R/NO-ELIP were observed to be relatively fast, with a release time of approximately one hour, while the release of encapsulated rosiglitazone was slower. NO-ELIP administration can inhibit intimal proliferation after acute artery injury by suppressing ICAM expression through the NF-κB-dependent pathway in human endothelial cells [35]. Future investigations are needed to explore the potential synergistic effects of rosiglitazone and NO on VSMC proliferation and inflammation following angioplasty. Further elucidation of the molecular mechanism underlying the cardiovascular-protective effect of PPARγ and NO could provide valuable mechanistic insights.

Clinical trials and post-market surveillance studies have established the effectiveness of PPARγ agonists in preventing atherosclerosis progression. However, such agents have been associated with adverse effects, notably PPARγ agonists. Recently, the FDA raised concerns regarding the systemic toxic effects of rosiglitazone [4]. In this study, we used rosiglitazone as a prototype to demonstrate the effectiveness of our developed delivery platform. Our platform is highly versatile and can be applied to other hydrophobic therapeutics, including pioglitazone.

Following local administration, positively-charged liposomes can adhere to the endothelial surface. Future studies will employ antibody conjugation for targeted delivery. Therefore, this platform integrates a bioactive gas and a therapeutic agent within a single moiety, allowing specific targeted delivery with high local concentrations and minimal systemic toxicity.

We have demonstrated the dual functionality of our nanoparticles as a contrast agent for ultrasound imaging and a carrier for targeted drug delivery. This novel drug delivery strategy suggests a promising approach to delivering therapeutic agents to the vascular walls with active atheroma. In clinical practice, current drug combinations are typically administered separately. Here, we present a liposomal codelivery system that effectively encapsulates two drugs for therapeutic delivery. This system offers the advantage of “synchronizing” the distribution and pharmacokinetic properties of the combined drugs, providing a more seamless transition from in vitro to in vivo settings. Furthermore, due to the inherent acoustic activity of the liposomal codelivery platform, it can potentially be activated by ultrasound [36], which, when applied to the disease site, can break down the liposomes and enhance local delivery and penetration. The potential of ultrasound in controlling therapeutic release from our liposomal platform is a promising avenue for future research.

## 5. Conclusions

The aim of this study was to develop a platform for the co-encapsulation and co-delivery of therapeutics, focusing specifically on investigating the delivery of NO and a PPAR agonist. Co-delivery of rosiglitazone and NO demonstrated the most effective approach in inhibiting intimal hyperplasia. This is attributed to the synergistic effects of NO and rosiglitazone on ICAM expression and VSMC proliferation, which enhance drug penetration into the vascular wall. Our study highlights the potential of the echogenic liposome carrier as a promising platform for enhanced therapeutic delivery.

## Figures and Tables

**Figure 1 biomolecules-13-00861-f001:**
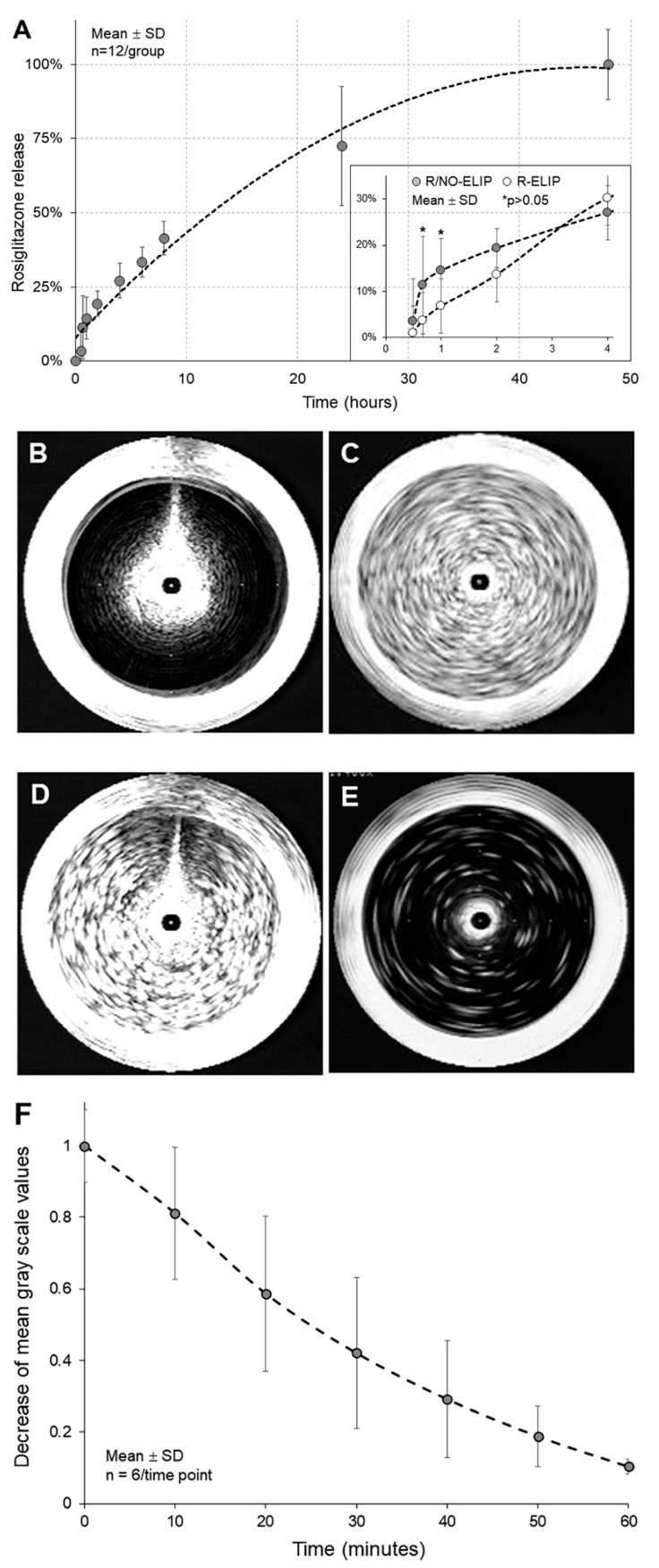
Release profiles of rosiglitazone and NO from R/NO-ELIP. (**A**) Quantification of released rosiglitazone assessed by measuring fluorescence intensity (EX317nm/EM372nm). Gas-induced ultrasound reflectivity of liposomes determined by intravascular ultrasound imaging of (**B**) conventional liposomes, (**C**) NO-ELIP, (**D**) R/NO-ELIP prior to release, and (**E**) R/NO-ELIP at 60 min after NO release. (**F**) NO release profiles determined by ultrasound imaging of R/NO-ELIP.

**Figure 2 biomolecules-13-00861-f002:**
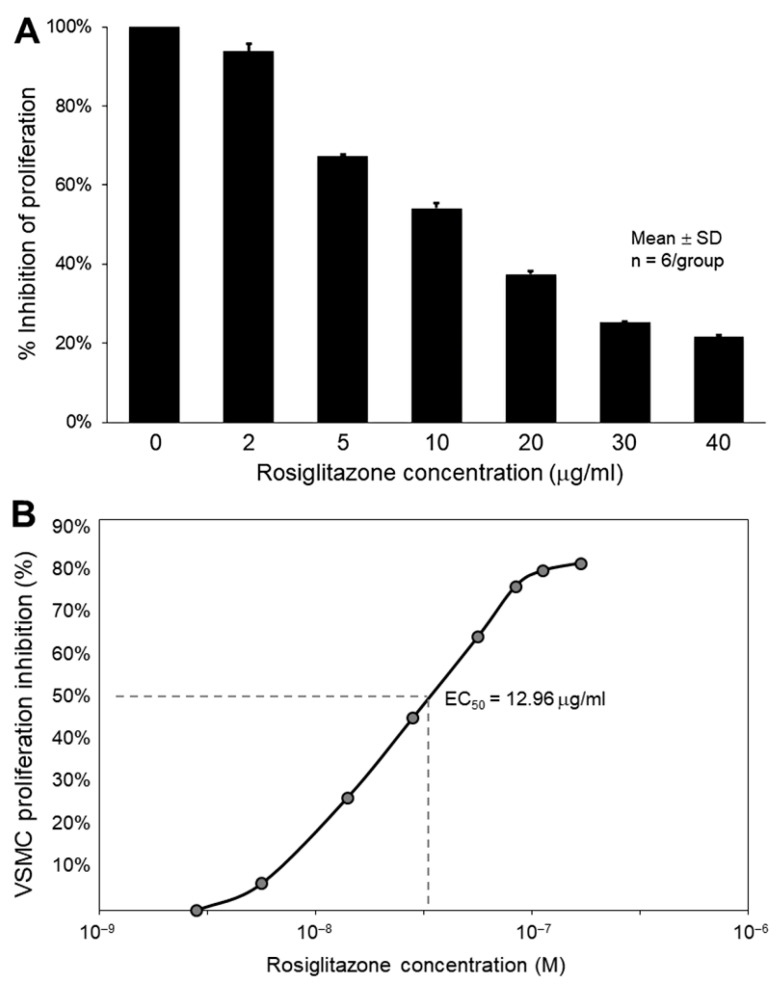
Effect of R/NO-ELIP on VSMC proliferation after 24 h of treatment. (**A**) A dose-dependent inhibition of serum-stimulated VSMC growth by rosiglitazone treatment (**B**) with EC50 at 12.96 μg/mL.

**Figure 3 biomolecules-13-00861-f003:**
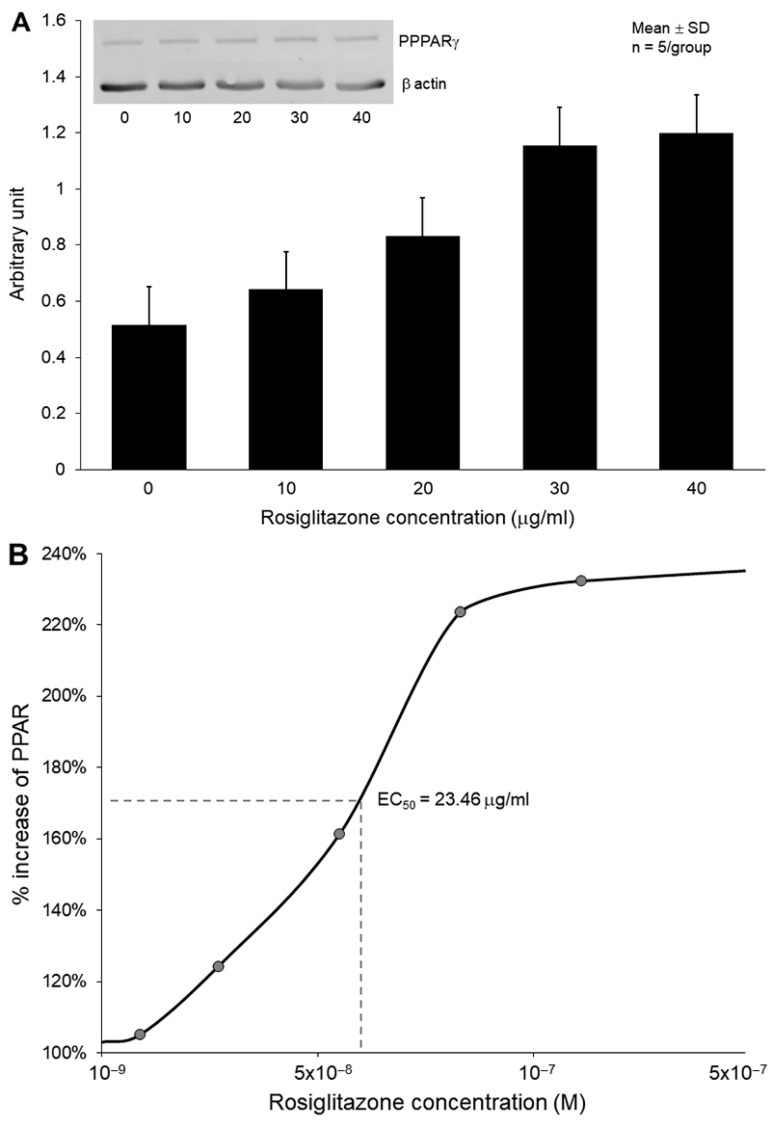
Effect of R/NO-ELIP on HUVECs. (**A**) Rosiglitazone treatment exhibited a dose-dependent increase in PPAR phosphorylation expression (**B**) with EC50 at 23.46 μg/mL.

**Figure 4 biomolecules-13-00861-f004:**
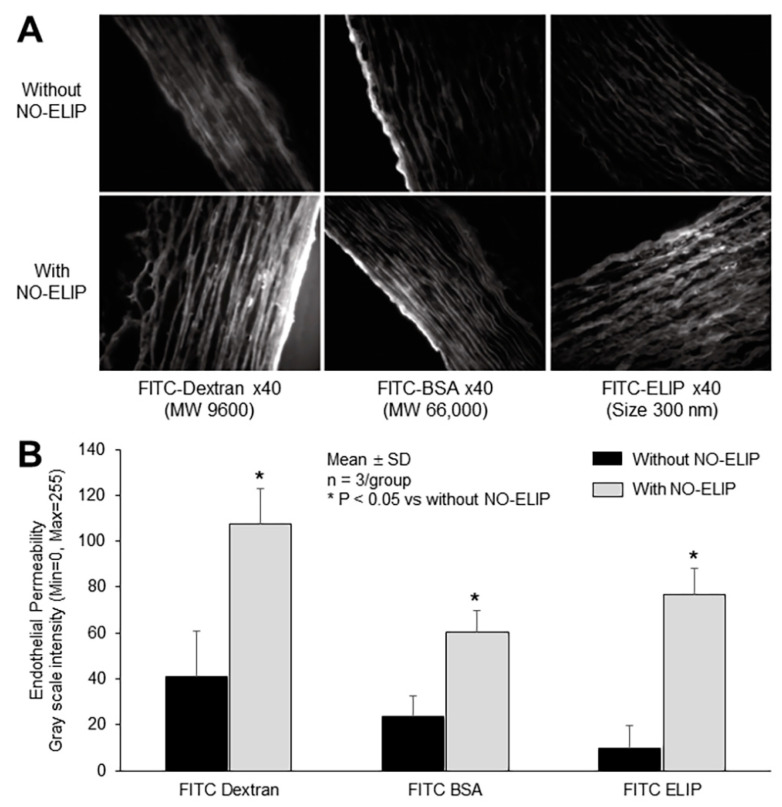
(**A**) Facilitated penetration of compounds of varying sizes into the vascular wall by NO-ELIP treatment demonstrated by fluorescence microscopy and (**B**) quantitation.

**Figure 5 biomolecules-13-00861-f005:**
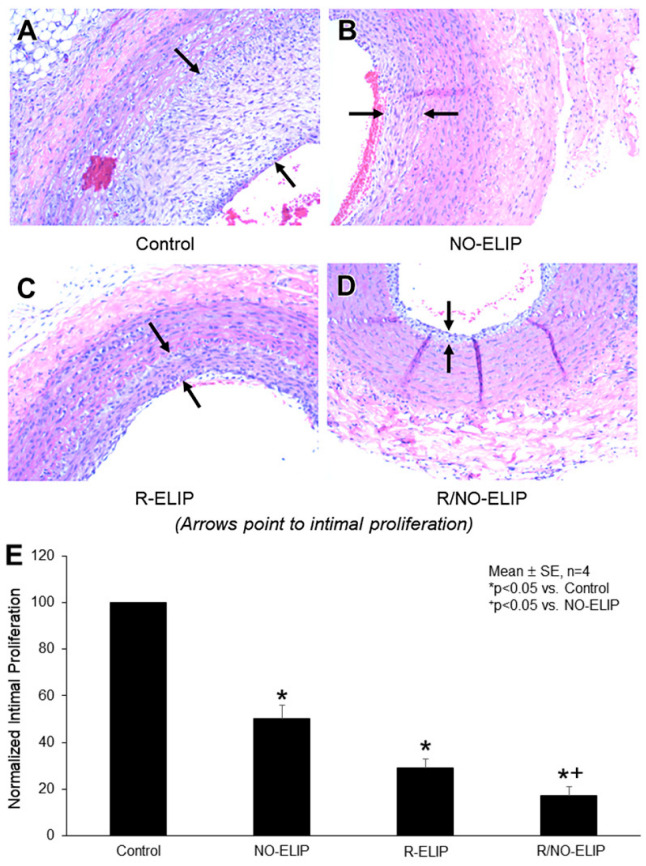
Impact of local delivery of R/NO-ELIP on neointimal hyperplasia following balloon-injured injury to the arteries. Representative histological sections (H&E × 400) of the common carotid artery 14 days post-injury, (**A**) with no treatment, (**B**) treated with NO-ELIP, (**C**) treated with R-ELIP, and (**D**) treated with R/NO-ELIP. (**E**) Quantitative evaluation of the intima/media thickness ratio.

**Figure 6 biomolecules-13-00861-f006:**
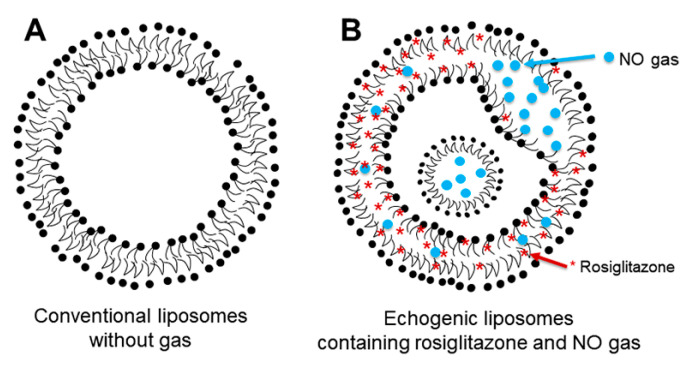
Schematics of the liposomal platform designed for co-encapsulation and delivery of therapeutic gas and therapeutic agents. (**A**) The conventional liposome with lipid bilayers comprising hydrophilic and hydrophobic cores. (**B**) Echogenic liposome allowing the co-encapsulation of therapeutic gas and therapeutic agents. Upon encapsulation, NO behaves similarly to hydrophobic drugs, residing either between the two monolayers of the liposome bilayer or within the bilayer of liposomes for energetic reasons. When co-encapsulated with bioactive gas, hydrophobic therapeutics such as rosiglitazone remain in the lipid bilayer.

## Data Availability

The data presented in this study are available on request from the corresponding author. All data generated or analyzed during this study are included in the manuscript.

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
