# Peer review of "Co-Delivery of Therapeutics and Bioactive Gas Using a Novel Liposomal Platform for Enhanced Treatment of Acute Arterial Injury"

_biomolecules, 2023, doi:10.3390/biom13050861_

Round 1

Reviewer 1 Report

In this study, the authors developed an echogenic liposome encapsulating NO and rosiglitazone for the treatment of acute arterial injury. The developed liposome could achieve a sustained release of rosiglitazone over 90 h and NO release over 60 min. In vitro and ex vivo studies showed that the NO/rosiglitazone loaded liposomes could inhibit VSMC proliferation, suppress endothelial inflammation, and increase endothelial permeability. In vivo studies on rabbit balloon-induced injury model showed that the double-loaded liposomes significantly reduced neointimal hyperplasia after local administration. The concept of co-delivery of bioactive gas and small molecule drug using liposomes is novel, and results are promising. Some questions might need to be addressed before the manuscript is accepted for publication.

1. There seems to be a huge difference between the release kinetics of NO (60 min) and rosiglitazone (> 80 h), which leads to several questions.

a. In section 2.2, NO-ELIP, R-ELIP, and R/NO-ELIP were subject to the dialysis study. However, only the result of R/NO-ELIP was shown. It would be useful to compare the release kinetics of R-ELIP and R/NO-ELIP to determine the effects of NO to rosiglitazone release kinetics.

b. The release of rosiglitazone was determined by the dialysis method. What are the material and COMW of the dialysis tube used in the experiment? Many studies have suggested that in dialysis-based drug release assays, the permeation of the drug through the dialysis membrane can be the rate-limiting step, leading to underestimated drug release rate. Determining how fast the free drug permeates through dialysis tubes would be valuable for method validation purposes.

2. While R/NO-ELIP showed greater inhibition effects in VSMC proliferation and endothelial inflammation, the mechanisms of synergistic effects of NO and rosiglitazone are not discussed in depth, and the proposed mechanisms are not supported by experimental results. For example, In line 339, it was suggested the reduction of neointimal hyperplasia was caused by the activation of PKG by rosiglitazone and NO. However, the PKG levels were not determined in the experiments. Similar to the statement in line 352 on the synergistic effect between NO and rosiglitazone on NF-kB site, which is also not supported by experimental results. Additional experiments or literature reviews on the mechanisms of NO and rosiglitazone would be helpful for this study.

3. The treatment application of the developed liposome should be clarified. The title suggested the developed liposome was for the treatment of acute arterial injury. However, later in the discussion (line 362), it was also stated that the liposome could be used to mitigate the formation of early atherosclerosis lesions, which is far-reaching and not supported by experiment results. The functionality for ultrasound imaging (line 377) is also not proved experimentally since no in vivo ultrasound was performed. Thus, some revision might be needed to clarify the experimentally supported therapeutic application of the developed liposomes.

Reviewer 2 Report

Dear Editor

I have read with interest the manuscript presented by Dr. Huang, Dr. Kim and collaborators, on the co-delivery of therapeutics and bioactive gas via liposomes, against atherosclerosis. In this specific case, nitrix oxide (NO) and rosiglitazone (R) have been used as drugs for investigating a potential anti-atherosclerosis treatment. R is an agonist of a nuclear receptor (PPAR-gamma), which stimulate the production of NO synthase; but its systemic use is limited by side-effects (thus, the idea of R delivery via liposomes). NO was co-encapsulated in liposomes in order to increase the penetration of R into the vascular wall and at the same time acting as anti-inflammatory, anti-thrombotic and anti-proliferative agent.

Authors claim a synergic effect of NO and R.

The major problem in this article, which is technically well done, is about limiting the generality of sentences about synergy to only one of the three aspects investigated (the one in Figure 5), while for the other effect synergy is not demonstrated. I would suggest you, editor, to invite the Authors to tune down the sentences about synergy when referred to data in Figure 2 and Figure 3. Also, the use of mean +/- SE instead of mean +/- SD seem an opportunistic choice not to expose too much the variability of data, especially in Figure 2 and Figure 3. So, Authors can surely claim synergy, but only with respect to the effect depicted in Figure 5.

Major

  • sentence in section 3.2 about “the greatest reduction…” should be more carefully evaluated. Surely, the difference shown in Figure 2C (non-treated vs R/NO-ELIP) is statistically significant (P<0.05), but authors cannot state that R/NO-ELIP is more effective than R-ELIP or NO-ELIP. So, the “greatest reduction” could be obtained, in other independent experiments, with R-ELIP or NO-ELIP. In other words, the differences between R-ELIP, NO-ELIP and R/NO-ELIP does not seem statistically significant. If this is the case (and indeed it is), synergy cannot be claimed from this experiment alone.
  • Also, if NO-ELIP has similar effect than R-ELIP and NO is completely released in 1 hour, while R is released in 9 hours, it means that R to show very moderate bioactivity when compared to NO, using the reported amounts. i.e., NO alone is sufficient to show the bio-activity, that is not statistical significantly different than R/NO combined.
  • similar comment (as in Figure 2C) can be done about Figure 3C. Statistical significance is vs non-treated samples.
  • Synergy is instead demonstrated by data in Figure 5, i.e., with respect to a specific feature (impact on neointimal hyperplasia)
  • Discussion should be then corrected, e.g., lines 325-332, highlighting the cases in which synergy has been demonstrated (statistically) and where hasn’t.

Minor

  • Line 85 (and elsewhere): please render “sn” in italics
  • Line 93: please be more specific, as I understand only later that R precipitates and liposomes stay in suspension, not the contrary (indeed, the supernatant (line 94) contained liposomes (?)). But see comment below
  • Equation 1: but now I do not understand equation 1, because F total – F supernatant should then correspond to the F of precipitated R, so the calculated % seems to correspond to free (unencapsulated) R. Please check it. The whole situation is not clear. Please state clearly where is located R in all steps of the described procedure.
  • Section 2.5 and following: Authors need to specify better what does it mean, e.g., 10 ug/mL liposomes. Is “10” referred to R or to lipids or….? And how it has been calculated. If it refers to R only, then, what does it mean “NO-ELIP” 20ug/mL? If it refers to lipids, actually authors have never quantified lipids after the initial centrifugation (line 93) – I guess. Please explain.
  • In section 3.1, explain better what you mean as 10 ul NO/argon.
  • emphasis should be added to explain that NO is released within 1 h, while R is released within 90 hours
  • Figure 2 caption: please add that treatment took 24 h as also specified in section 2.4, and figure 3 too.
  • Discussion (line 322): Figure 6 is missing
  •  

Round 2

Reviewer 1 Report

The authors have addressed the questions and comments in the revised manuscript. 

Reviewer 2 Report

the Authors have clarified several points about liposome preparation, deleted two figures, and tune down the claims about synergic effect (limited only to the previous Figure 5).

I suggest to accept the current manuscript.